# Prevalence and Risk Factors for Cognitive Frailty in Aging Hypertensive Patients in China

**DOI:** 10.3390/brainsci11081018

**Published:** 2021-07-30

**Authors:** Can Wang, Jiechun Zhang, Chengping Hu, Yanbo Wang

**Affiliations:** 1Clinical Research Center for Mental Disorders, Shanghai Pudong New Area Mental Health Center, School of Medicine, Tongji University, 165 Sanlin Road, Shanghai 200124, China; 13917168835@163.com (C.W.); Zhangjch@shspdjw.com (J.Z.); 2Division of Medical Humanities and Behavioral Sciences, Tongji University School of Medicine, 1239 Siping Road, Shanghai 200092, China

**Keywords:** cognitive frailty, aged, hypertension, prevalence

## Abstract

Hypertension is one of the most common chronic diseases and a major risk factor for stroke, myocardial infarction and cardiovascular death. Cognitive frailty is an important predictor of all-cause mortality and dementia in aging individuals. Hypertension is closely related to cognitive frailty and these two conditions often coexist in aging individuals. Few studies have explored the relationship between hypertension and cognitive frailty in the Chinese population. This study investigates the epidemiological characteristics of and factors related to cognitive frailty in aging Chinese patients with hypertension. In total, cognitive function, weakness, social support, depression and sociodemographic were assessed in 305 participants aged 60 and over. Univariate and multivariate logistic regression models were constructed. The prevalence of cognitive frailty in aging Chinese hypertensive patients was 9.8% (95% CI = 6.4–13.2%). After adjusting for confounding variables, logistic regression showed that the course of hypertension (6–10 years, OR = 8.588, 95% CI = 1.608–45.859;course of more than 10 years, OR = 9.020, 95%CI = 1.854–43.892), multimorbidity (OR = 11.231, 95% CI = 2.912–43.322), depression (OR = 6.917, 95% CI = 2.424–19.738) and social support (OR = 0.187, 95% CI = 0.071–0.492) were independently associated with cognitive frailty. The prevalence of cognitive frailty in aging patients with hypertension in China should not be ignored. The course of hypertension, multimorbidity and depression are the risk factors of cognitive frailty in the aging population and a better level of social support is the protective factor for cognitive frailty.

## 1. Introduction

According to the World Health Organization, the number of people over the age of 60 years is expected to reach 2 billion by 2050. The aging of the population has led to a significant increase in age-related diseases. Frailty is one of the most urgent challenges in the aging population. Frailty is a multidimensional syndrome that can lead to declines in physical, cognitive, psychological and social functioning [1]. Studies have found that frailty and cognitive impairment may have a common pathophysiological mechanism [2,3]. The two interact to accelerate the declines in physical function and cognitive function. Therefore, the concept of cognitive frailty (CF) was proposed and has gradually attracted attention. CF is a type of frailty in patients who do not have dementia but have physical frailty (PF) and mild cognitive impairment (MCI) [4]. Researchers have found that, compared with the consideration of frailty and cognitive impairment, the consideration of CF can improve the prediction of disability, falls, dysfunction and death in the aging population [5,6,7,8,9]. CF is reversible, so it may become a target for the prevention of disability [10].

The prevalence of hypertension in people aged 60 years and older is 20.1% [11]. Hypertension is another urgent health problem in the aging population. The inducing and influencing factors of hypertension include biological, social and psychological aspects [12]. On the basis of drug treatment, experts advocate a healthy lifestyle and correct stress-coping strategies to prevent hypertension [13,14,15,16,17]. Previous studies have shown that hypertension is a common risk factor for physical frailty and cognitive decline [18]. High blood pressure can damage brain capillary [19], resulting in a decline in cognitive function and an acceleration in the development of dementia. That seems to have become a consensus, but a systematic review shows that the adverse effects of hypertension on cognitive function depend on age. In middle-aged subjects (40–64 years old), the increase of blood pressure is positively correlated with cognitive impairment, but this correlation decreases with age. Even in those aged 75 and over, hypertension seems to have a protective effect on cognitive function [20].In addition, frailty is common in patients with hypertension [21,22] and is a strong predictor of mortality, hospitalization and falls [23]. There have been studies on hypertension in individuals with frailty and cognitive decline. Given that the combined effects of frailty and cognitive decline are cumulative [24], it is necessary to screen for cognitive decline in patients with hypertension. Shimada H et al. [25] investigated chronic diseases in individuals with CF, and the most common chronic disease was hypertension (56.1%). The difference in the prevalence of hypertension between the aging patients with and without CF was significant (*p* < 0.001). However, there has not been a study specifically focused on screening for CF in aging hypertensive individuals. The purpose of this study is to investigate the prevalence of CF in the aging with hypertension in China and explore its possible influencing factors, in order to provide new targets for the prevention and treatment of CF and its adverse consequences in hypertensive aging individuals.

## 2. Materials and Methods

### 2.1. Research Design and Subjects

This cross-sectional study was conducted in the Pudong New Area of Shanghai from 1 December 2020 to 31 March 2021. With the assistance of neighborhood committee staff, we recruited participants who met the following inclusion criteria from the aging living in the community by telephone: (1) 60–89 years old; (2) Receive anti-hypertensive treatment or have a history of hypertension. People with dementia and serious mental disorders were excluded. We contacted a total of 343 aging people, of whom 28 refused to participate in the study because of “limited physical strength” and “no interest”. Ten did not complete the test because of lack of patience or being interrupted by telephone. Finally, 305 aging people were included in the study. The research protocol was approved by the ethics committee of Pudong New Area Mental Health Center (PDJWMLL2020025). We fully explained the research process to all participants and obtained their written informed consent.

### 2.2. Sociodemographic Characteristics

The sociodemographic characteristics included sex, age, course of hypertension, marital status, multimorbidity, number of drugs used, body mass index (BMI), smoking, alcohol consumption and physical exercise. The age groups were 60–69 years old, 70–79 years old and 80–89 years old. The course of hypertension was divided into 1–5 years, 6–10 years and >10 years. The diagnosis and duration of hypertension were collected from patients’ self-reports and the medical insurance records. Marital status was divided into unmarried, married, divorced and widowed. Multimorbidity was categorized as “yes” and “no” according to whether they had two or more chronic diseases. According to the number of drugs taken, they were divided into 1, 2, 3 and 4 groups, which respectively represented 1 drug, 2 drugs, 3 drugs and 4 or more drugs. BMI was divided into underweight (<18.5 kg/m^2^), normal (18.5–24.9 kg/m^2^) and overweight and above (≥25.0 kg/m^2^). By asking the following questions: “Have you ever smoked?”, “Do you have a history of drinking?”, “Do you exercise more than two hours a week?”, smoking, alcohol consumption and physical exercise were divided into binary responses (“yes” or “no”). Chronic diseases included cardiovascular diseases, such as hypertension and coronary heart disease; lung diseases; gastrointestinal diseases; kidney diseases; prostate diseases; thyroid diseases; eye diseases; ear diseases; osteoporosis; arthritis; etc.

### 2.3. Cognitive Frailty

CF refers to physical frailty (PF) and mild cognitive impairment MCI (excluding dementia) in aging individuals. We used the Mini Mental State Examination (MMSE) [26] and Frailty Phenotype (FP) [27] to measure CF. The MMSE [26] is used to evaluate the cognitive function of individuals. The score ranges between 0 and 30. The higher the score is, the better the cognitive function. The score used to diagnose MCI is stratified by education levels: illiterate (≤17), primary school (≤20), junior high school (≤22) and senior high school and above (≤24). The frailty phenotype [27] is used to evaluate PF and includes five physiological indicators: involuntary weight loss, fatigue, reduction in grip strength, reduction in walking speed and reduction in physical activity. The total score ranges from 0 to 5. A score of 0 indicates no frailty, a score of 1 or 2 indicates the prophase of frailty and a score from 3 to 5 indicates frailty.

### 2.4. Social Support

The social support rating scale (SSRS) [28] was used to measure social support. It includes three dimensions: objective support (three items), subjective support (four items) and support utilization (three items). The score for objective support ranges from 4 to 16, the score for subjective support ranges from 5 to 38 and the score for support utilization ranges from 3 to 12. The higher the total score is, the higher the support level. The reliability and validity of the SSRS scale was good (Cronbach’s α coefficient = 0.941).

### 2.5. Depression

The Geriatric Depression Scale (GDS-15) was used to evaluate the depression status of the aging participants. The GDS-15 is a simplified version of the GDS and was revised in 1986 [29]. There are 15 items in total. Each item is scored as 1 or 0 according to whether the response is “yes” or “no”, respectively, and negative questions are scored in reverse. The total score ranges from 0 to 15. The higher the score is, the higher the level of depression. A score from 0–4 indicates no depression, 5–7 indicates mild depression, 8–11 indicates moderate depression and 12–15 indicates severe depression. The sensitivity and specificity of the GDS-15 were 0.96 and 0.88, respectively. The α coefficient was 0.82.

### 2.6. Data Analysis

Excel was used to establish the database with double data entry, and SPSS 26.0 (Windows version) from IBM was used to describe and analyze the data. The Kolmogorov-Smirnov test was used to test the normality of the distributions of continuous variables. The measurement data conformed to normal distributions and the means ± standard deviations are used to describe these variables. The count data are expressed as frequencies (percentages). The chi-square test was used to determine the differences between groups in the demographic and clinical characteristics, and logistic regression was used to evaluate the factors influencing cognitive frailty. All statistical tests were two-sided, with *p* less than 0.05 indicating statistical significance.

## 3. Results

There were 30 cases of CF in 305 aging patients with hypertension, and the prevalence rate was 9.8%. Other demographic data (sex, age, course of hypertension, marital status, multimorbidity, number of drugs used, BMI, smoking, alcohol consumption and physical exercise) are shown in Table 1. The univariate analyses of CF in aging hypertensive patients (see Table 1) showed that there were no significant differences in the prevalence of CF among aging hypertensive patients stratified by sex, age, marital status, number of medications, BMI, smoking status and alcohol consumption status (*p* > 0.05). There were significant differences in the incidence of CF among aging hypertensive patients stratified by the duration of hypertension, social support level, physical exercise, multimorbidity and depression (*p* < 0.05).

The presence of CF was the dependent variable. The influencing factors (course of hypertension, multimorbidity, physical exercise, social support, depression) identified in the univariate analyses were included as the independent variables. Logistic regression analysis with backward selection (likelihood ratio) was performed. The inclusion criterion was 0.05 and the exclusion criterion was 0.10. Participants who had hypertension for 6–10 years and more than 10 years are 8.588 times (95% CI = 1.608–45.859) and 9.020 times (95% CI = 1.854–43.892) more likely to have CF than those with a course of less than 5 years; multimorbidity (OR = 11.231, 95%CI = 2.912–43.322) and depression (OR = 6.917, 95%CI = 2.424–19.738) are independent risk factors for cognitive frailty; better social support (OR = 0.187, 95% CI = 0.071–0.492) is a protective factor for cognitive frailty (see Table 2).

## 4. Discussion

CF is a serious threat to the quality of life of aging individuals. Although a number of epidemiological surveys on the prevalence of CF in the aging population have been carried out, there is no relevant research in the population of aging patients with hypertension. To our knowledge, this is the first report on the prevalence of CF in aging hypertensive patients in China. The results of this study can provide a reference to facilitate the development of intervention strategies to improve CF in aging patients with hypertension.

In this study, the prevalence of CF in aging individuals with hypertension was 9.8% (95% CI = 6.4–13.2%), which was higher than the results reported in several studies performed in China and elsewhere of 2.6% [30], 6.6% [31], 7.1% [32] and 3.2% [33]. Hypertension can lead to declines in attention, executive function and other cognitive functions [34]; damage the cardiovascular system; increase physical vulnerability, and accelerate the development of frailty. Therefore, the prevalence of CF is relatively higher in aging individuals with hypertension, which is consistent with the results reported by Arai H et al. [35]. That study showed that the prevalence of CF was higher in patients with chronic debilitating diseases than in community-dwelling aging individuals. The latest survey in Brazil [36] used the MMSE and instrumental activities of daily living (IADL) to screen for CF and found a prevalence rate of 10.9%, which was higher than that in this study. This may be due to the use of different assessment tools and geographical differences. A cross-sectional survey conducted in China [37] found that the prevalence rate of CF was 23.08%. The higher rate in that study may be because the aging individuals were not recruited from the community but from among patients with cerebrovascular disease in the Department of Neurology. Those aging people had a heavy burden of disease, leading to a high prevalence rate of CF. A longitudinal survey of 705 aging Chinese people over 90 years old [6] showed that the prevalence rate of CF was 50.2%, which may be related to the high average age (93.6 ± 3 years). At present, the reported prevalence of CF has a wide range, which is due to the use of different assessment tools and diagnostic criteria, as well as the differences in the level of economic development and other factors. At present, there is no unified assessment tool for CF. Researchers often use the presence of a decline in physical function combined with the scores on cognitive function assessment scales to screen for CF. Standardized assessment tools need to be developed.

This study found that the risk of CF in hypertensive aging individuals is related to the course of hypertension but not age, which was consistent with the results of the study by Power et al. [19]. That longitudinal study, which investigated the relationship between the course of hypertension and cognitive function in 758 people, confirmed that the average cognitive score decreased by 0.02 standard units per year with an increase in the course of hypertension, regardless of the age at onset. Chronological age cannot fully reflect the differences in health status in patients with hypertension, and CF is closely related to biological age. There is a consensus that hypertension can damage the nervous system and that damage is persistent and cumulative. Therefore, the early identification, prevention and control of hypertension is needed to prevent and reverse CF. Our results may provide a beneficial reference for the prevention and treatment of cognitive decline in the aging with hypertension in China.

In this study, 122 people (40%) had two or more chronic diseases and multimorbidity was an independent risk factor for CF (OR = 11.231, 96% CI = 2.912–43.322), which was in accord with the results of several studies [38,39,40]. Previous studies have shown that multiple diseases generally coexist in aging individuals and that multimorbidity is related to higher incidences of physical frailty and cognitive decline [41,42,43]. Multimorbidity may reflect age-related multiple organ and organ system failure, which reduces the body’s resistance and tolerance and promotes the occurrence of physical frailty. The combination of cerebrovascular disease, such as hypertension, and other chronic diseases has a synergistic effect, accelerating the development of CF. Therefore, to promote the overall health of the aging population and prevent CF, clinicians should comprehensively evaluate the disease burden in patients before formulating the treatment and nursing plan.

Our study shows that depression affects the development of CF in aging community-dwelling individuals with hypertension. Aging individuals with depression are more prone to developing CF (OR = 6.917, 95% CI = 2.424–19.738), which is consistent with the findings of other studies [25,40,44]. Previous studies confirmed that there is a strong correlation between hypertension and depression in aging individuals and that the two are causal and interact with each other [45]. Solfrizzi V et al. [46] conducted a cross-sectional survey involving 2150 aging people aged 65 years and older in Italy and found that compared with the nonreversible CF group, the reversible CF group was significantly more likely to have depressive symptoms *(p* < 0.05). Ma et al. [47] investigated 5708 aging people without dementia in seven representative cities in China, and found that the prevalence rate of CF in the depression group was 15.1%, which was three times of that in the whole sample (3.3%). Malek Rivan NF et al. [48] found that depression (OR = 1.49, 95% CI = 1.34–1.65, *p* < 0.001) was a risk factor for CF in aging community-dwelling individuals. Depression can lead to the upregulation of tumor necrosis factor (TNF) and interleukin (IL)-6. Inflammatory cytokines can penetrate the blood–brain barrier, act on skeletal muscle, accelerate the occurrence of sarcopenia and cognitive dysfunction, and eventually lead to CF [49]. Depression in the aging population is characterized by less verbalization, less movement, mental fatigue, unwillingness to communicate with others, and the continuous lack of physical activity and interpersonal communication, leading to cognitive decline and physical frailty [49].

Social support refers to the spiritual support and material help that individuals perceive themselves as receiving through social contact and participation in social activities, which can relieve mental tension, relieve psychological stress and improve social adaptability. We found better social support was a protective factor for cognitive frailty (OR = 0.187, 95% CI = 0.071–0.492). Due to the limitations on activity and social interactions, frailty reduces the availability of social support. In addition, an individual’s social ability is based on their degree of cognitive function, and cognitive decline in aging individuals leads to a reduction in their interpersonal skills, resulting in less social support. The results of the study by Liu et al. [43] are consistent with the results of this study. He found that better social support can meet the physical and mental needs of aging individuals and reduce the incidence of frailty. Although there are still controversies regarding the relationship between social support and frailty in the aging population [50], social support can effectively reduce depression in aging individuals with hypertension [50,51], as mentioned above, and depression is an independent risk factor for CF. Therefore, enhancing the social support received by aging individuals and helping them make better use of the social support they receive [52] will help reduce depression and maintain the physical and mental health of this population, preventing the development of CF.

This was a cross-sectional study and the results were based on the analysis of baseline data. Further follow-up studies are needed to investigate the incidence and possible predictors of CF. Information about the participants with chronic diseases was collected through self-report and from medical insurance cards. We could not confirm the self-reported information. In addition, we did not measure blood pressure for participants, so we were unable to evaluate the relationship between different blood pressure levels and cognitive frailty in the aging population.

## 5. Conclusions

The prevalence of CF is high in the aging Chinese hypertensive population. We should comprehensively evaluate aging individuals and develop multimodal personalized interventions. Aging individuals with a history of hypertension longer than 5 years and multiple comorbid diseases are at higher risk of CF. Reducing depression and improving the level of social support can reduce the prevalence of CF in the aging population.

## Figures and Tables

**Table 1 brainsci-11-01018-t001:** General demographic data and univariate analyses.

Variable	Number	Non-CF (n = 275, 90.2%)	CF (n = 30, 9.8%)	Statistical Values	*p*-Values
Sex				0.913	0.339
Male	108	95 (34.5%)	13 (43.3%)		
Female	197	180 (65.5%)	17 (56.7%)		
Age group				3.640	0.162
60–69	140	130 (47.3%)	10 (33.3%)		
70–79	114	98 (35.6%)	16 (53.3%)		
80–89	51	47 (17.1%)	4 (13.3%)		
Course ※				17.378	<0.001
1–5 years	119	117 (42.5%)	2 (6.7%)		
6–10 years	82	72 (26.2%)	10 (33.3%)		
>10 years	104	86 (31.3%)	18 (60.0%)		
Marital status				4.332	0.228
Unmarried	15	13 (4.7%)	2 (6.7%)		
Married	243	223 (81.1%)	20 (66.7%)		
Divorced	2	2 (0.7%)	0 (0.0%)		
Widowed	45	37 (13.5%)	8 (26.7%)		
Multimorbidity				12.477	<0.001
Yes	122	119 (43.3%)	3 (10.0%)		
No	183	156 (56.7%)	27 (90.0%)		
Number of drugs				6.480	0.066
1	170	158 (57.5%)	12 (40.0%)		
2	80	70 (25.5%)	10 (33.3%)		
3	40	36 (13.1%)	4 (13.3%)		
4	15	11 (4.0%)	4 (13.3%)		
BMI				1.167	0.558
<18.5 kg/m^2^	126	113 (41.1%)	13 (43.3%)		
18.5–24.9 kg/m^2^	35	32 (11.6%)	3 (10.0%)		
≥25 kg/m^2^	144	130 (47.3%)	14 (46.7%)		
Smoking				0.000	1.000
Yes	18	16 (5.8%)	2 (6.7%)		
No	287	259 (94.2%)	28 (93.3%)		
Alcohol					
consumption				1.263	0.261
Yes	65	61 (22.2%)	4 (13.3%)		
No	240	214 (77.8%)	26 (86.7%)		
Physical exercise				6.594	0.010
Yes	254	234 (85.1%)	20 (66.7%)		
No	51	41 (14.9%)	10 (33.3%)		
Depression				34.434	<0.001
Yes	273	256 (93.1%)	17 (56.7%)		
No	32	19 (6.9%)	13 (43.3%)		
Social support				24.637	<0.001
Low level	59	43 (15.6%)	16 (53.3%)		
Medium/high level	246	232 (84.4%)	14 (46.7%)		

Note: ※ refers to the course of hypertension.

**Table 2 brainsci-11-01018-t002:** Backward selection (likelihood ratio) logistic regression analysis of cognitive frailty in aging patients with hypertension.

Variable	B	S.E.	Wald’s	*p*-Values	OR	95%CI
Course of Hypertension						
1–5 years					1	
6–10 years	2.150	0.855	6.330	0.012	8.588	1.608–45.859
>10 years	2.199	0.807	7.422	0.006	9.020	1.854–43.892
Multimorbidity						
No					1	
Yes	2.419	0.689	12.332	0.000	11.231	2.912–43.322
Depression						
No					1	
Yes	1.934	0.535	13.069	0.000	6.917	2.424–19.738
Social support						
Low level					1	
Medium/high level	−1.677	0.493	11.553	0.001	0.187	0.071–0.492

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
