# Peer review of "Prevalence and Risk Factors for Cognitive Frailty in Aging Hypertensive Patients in China"

_brainsci, 2021, doi:10.3390/brainsci11081018_

Round 1

Reviewer 1 Report

Summary: The current manuscript investigated cognitive frailty in elderly hypertensive patients in China.

Although the authors present interesting, clinically findings, some aspects could be improved.

Please consider the following suggestion for revision:

Introduction: Overall, the introduction provides a brief background and rationale for the research. 

This choice does not allow to analyze the problem fully, and it may be useful to mention other psychological aspects related to high blood pressure, such as alexithymia, anxiety, depression, coping strategies. I suggest the reading of the following articles: “Casagrande, M., Forte, G., Guarino, A., Favieri, F., Boncompagni, I., Germanò, R., Germanò, G., Mingarelli, A. (2019). Alexithymia: A Facet of Uncontrolled Hypertension. International Journal of Psychophysiology, 146, 180-189, DOI: 10.1016/j.ijpsycho.2019.09.006” and “Casagrande, M., Boncompagni I., Mingarelli A., Favieri F., Forte G., Germanò R., & Germano G., Guarino, A. (2019) Coping styles in individuals with hypertension of varying severity. Stress & Health, 35, 1-9, DOI: 10.1002/smi.288.”.

Furthermore, some review related to relationship between hypertension and cognitive function could be reported to better clarify the relationship ( for example  Forte, G., & Casagrande, M. (2020). Effects of Blood Pressure on Cognitive Performance in Aging: A Systematic Review. Brain Sciences, 10(12), 919.)

Methods: The method is succinct, but some improvements should be made to make it more comprehensive. Office blood pressure is considered? It is not clear.

Analysis: the analyses are well conducted. I appreciate the Tables.  

Results: The summary of the stud is well-defined and fits the planned analyses provide.

Discussion: This section could be improved, emphasizing the novelty of this research. Limitations section could be clarified.  

General comment: I would also encourage the authors to check all references

Reviewer 2 Report

The manuscript titled “Prevalence and risk factors for cognitive frailty in elderly hypertensive patients in China” proposed an epidemiological investigation on the prevalence and risk factors of cognitive frailty (CF) in elderly people suffering from hypertension in China. A cohort of 305 participants over 60 years were recruited and administered tests for measuring socio-demographic information, cognitive functioning, weakness, social support, and depression. Data were analyzed through logistic regressions. Results showed a prevalence of CF in elderly people with hypertension of 9.8%. Logistic regression models showed that the course of hypertension, multimorbidity, depression, and social support were associated with CF. Authors discussed their results in light of previous literature as well as the limitations of the study and gave hints for future research.

I carefully read the manuscript, and I think it may be of interest for the readers of Brain Sciences. However, I also think that several major points hamper the publication of the manuscript as a research article. Below there are my comments and suggestions.

Abstract section

Page 1 line 21: please clarify the meaning of OR1 and OR2. Moreover, several times throughout the manuscript this symbol “~” (tilde) is used to separate two numerical values, probably instead of this “-“ (dash). Please fix this issue.

Page 1 line 24: how can you say that “the prevalence of cognitive frailty is high in elderly Chinese patients…”? On what basis a prevalence estimate of 9.8% can be considered high?

Page 1 line 26: the last sentence of the abstract is a mere repetition of the results. Please try to outline a couple of lines of discussion of your results.

Introduction section

Page 1 line 39-41: the sentence beginning with “Researchers have found…” is unclear and difficult to understand. Please, try to clarify its meaning.

Page 2 lines 57-60: the first part of the sentence is the real aim of the manuscript, while the second part is not the aim, it may be a consequence of your results. I suggest removing the second part, since it’s not what you do in the manuscript, and save it for the discussion section. Maybe you could split the aim into two parts: 1) prevalence estimate and 2) risk factors.

Materials and Methods

Page 2 lines 65-67: Were those inclusion or exclusion criteria? Please specify and detail each inclusion and exclusion criterion.

Page 2 line 67: you wrote “A total of 305 elderly people were included in this study.” In my opinion, this statement is not sufficient to adequately describe the sampling process. How were the participants contacted, by which means? How many were contacted at first? How many refused to participate, and for what reasons? How many were at first included but then excluded due to your exclusion criteria? Please try to describe and better characterize the sampling process.

Page 2 lines 83-84: What were the exact questions (for example: "Have you ever smoked"?) regarding smoking, alcohol consumption and physical exercise asked to the participants?

Results section

Page 3 lines 127-128: please, calculate and report 95% CI for the prevalence estimate.

Page 3 lines 131-132: I would not use the word “incidence” as a synonym of “impact” in an epidemiological manuscript, since it has a precise meaning.

Page 3, line 137, Table 1: it is not clear what the column "Number of cases" represents. There are two groups: 1) Non-CF (n=275), and 2) CF (n=30), and I cannot understand the first column of numbers, which changes constantly through the different variables enlisted in the table.

Discussion section

Page 6 line 159: I think you meant “the prevalence” instead of “the incidence”.

Please check the manuscript for grammar issues and typos.

Reviewer 3 Report

General comments:

The paper reports on prevalence and risk factors of cognitive frailty in elderly hypertensive patients in China. Considering the prevalence of hypertension, the detrimental effects of hypertension on brain and peripheral blood vessels, inflammatory processes as a common feature of atherosclerosis, cerebrovascular disease, neurodegeneration and various processes contributing to frailty, this topic is of importance. The authors found a notable prevalence of cognitive frailty in patients with hypertension and a number of risk factors, some of which are amenable to intervention.

Specific issues:

  1. Section 2.2, Sociodemographic characteristics, line 82: BMI for overweight and above is given as “> 25”. Should not this read “≥ 25”? For normal BMI “18.5 – 24.9” is given, so “= 25” would not be defined. Please check.
  2. Section 2.4, Social support, line 101: No reference is provided for the social support rating scale (SSRS).
  3. Table 1, line 137: It is unclear what “Number of cases” as header of the second column means and what the numbers and percentages in the second column represent. In some lines, the numbers and percentages of columns three and four add up to the numbers and percentages of column two, in some lines they do not.
  4. Section 4, Discussion, lines 159 to 161: This was a cross-sectional study and, as reported in the Results section, the prevalence – not the incidence – of cognitive frailty was found to be 9.8%. The papers referred to as [21], [22], and [24] also report on prevalences, whereas the paper referred to as [23] reports on the incidence (7.1 per 100 person years, not 7.1%).
  5. Section 4, Discussion, line 185: Here it should read “Power et al. [12]” (rather than “Melinda et al. [12]”). The first author is Melinda Power, Melinda being the first name and Power the last name.
  6. Section 4, Discussion, line 238: Reference #42 (Nam et al. 2019) is not about cognitive frailty. Please check.

Round 2

Reviewer 2 Report

I carefully read the resubmitted version of the manuscript titled “Prevalence and risk factors for cognitive frailty in elderly hypertensive patients in China”. I think that the Authors did several changes according to referees’ comments, especially regarding introduction sections and results reporting, and that the present version has been improved a lot with respect to the previous one.

I have only a couple of minor remarks, which can be found below:

Page 3 lines 105-106: Please, report into the manuscript the questions regarding smoking, alcohol consumption and physical exercise

2) Table 1: I suppose that after the corrections the column “Number” now contains the sum of the columns “Non-CF” and “CF”. If so, the percentages in the brackets in the column “Number” should always sum up to 100% and can be removed. Please also check all the frequencies and all the percentages.
